# Light Quality Impacts Vertical Growth Rate, Phytochemical Yield and Cannabinoid Production Efficiency in *Cannabis sativa*

**DOI:** 10.3390/plants11212982

**Published:** 2022-11-04

**Authors:** Victorio Morello, Vincent Desaulniers Brousseau, Natalie Wu, Bo-Sen Wu, Sarah MacPherson, Mark Lefsrud

**Affiliations:** Department of Bioresource Engineering, McGill University, 21111 Lakeshore Road, Sainte-Anne-de-Bellevue, QC H9X 3V9, Canada

**Keywords:** THC, CBD, LED, HPS, light wavelength, terpenes

## Abstract

Light is one of the most crucial parameters for enclosed cannabis (*Cannabis sativa*) production, as it highly influences growth, secondary metabolite production, and operational costs. The objective of this study was to investigate and evaluate the impact of six light spectra on *C. sativa* (‘Babbas Erkle Cookies’ accession) growth traits and secondary metabolite (cannabinoid and terpene) profiles. The light spectra evaluated included blue (430 nm), red (630 nm), rose (430 + 630 nm, ratio 1:10), purple (430 + 630 nm, ratio 2:1), and amber (595 nm) LED treatments, in addition to a high-pressure sodium (HPS, amber-rich light) treatment as a control. All the LED light treatments had lower fresh mean inflorescence mass than the control (HPS, 133.59 g plant^−1^), and monochromatic blue light yielded the least fresh inflorescence mass (76.39 g plant^−1^). Measurement of Δ9-tetrahydrocannabinol (THC) concentration (%) and total yield (g plant^−1^) showed how inflorescence mass and THC concentration need to be analyzed conjointly. Blue treatment resulted in the highest THC concentration (10.17% m/m), yet the lowest THC concentration per plant (1.44 g plant^−1^). The highest THC concentration per plant was achieved with HPS (2.54 g plant^−1^). As with THC, blue light increased cannabigerol (CBG) and terpene concentration. Conversely, blue light had a lesser impact on cannabidiol (CBD) biosynthesis in this *C. sativa* chemotype. As the combined effects of the light spectrum on both growth traits and secondary metabolites have important ramifications for the industry, the inappropriate spectral design could cause a reduction in cannabinoid production (20–40%). These findings show promise in helping producers choose spectral designs that meet specific *C. sativa* production goals.

## 1. Introduction

Cannabis (*Cannabis sativa*) has been exploited as a medicinal plant for over two millennia [1]. With the global movement on cannabis legalization, *C. sativa* production has become one of the most rapidly expanding markets [2,3]. *C. sativa* synthesizes and accumulates more than 500 known secondary metabolites [4]. Considerable efforts are aimed at investigating this plant’s secondary metabolites, including cannabinoids and terpenes [5,6]. Major cannabinoids, including Δ9-tetrahydrocannabinol (THC) and cannabidiol (CBD), aid in reducing chronic pain, chemotherapy-induced nausea, vomiting, and improving multiple sclerosis spasticity symptoms [7,8,9].

Accumulated evidence suggests that botanical extracts containing cannabinoids, terpenes, and other secondary metabolites are more potent than isolated cannabinoids [10,11]. The synergistic action of *C. sativa* bioactive compounds, deemed the “entourage effect”, could provide unique therapeutic strategies promising new avenues in the treatment of pain, inflammation, depression, and multiple other disorders [12,13,14]. This focus on the specific cannabinoid-terpenoid ratio for the therapeutic use of the entourage effect means that cannabis breeders and growers must pay particular attention to horticultural practices that could help fine-tune secondary metabolite profiles in *C. sativa* [15].

Light is one of the crucial parameters for enclosed *C. sativa* production, as it greatly impacts the growth and its secondary metabolite accumulation of all plants through light intensity and spectra [16,17,18,19]. In the greenhouse industry, light-emitting diodes (LEDs) have been widely used for plant cultivation, as they are rated more energy-efficient over other conventional light sources, including high-pressure sodium (HPS) lamps [20,21]. Fixture efficacy plays an important role in *C. sativa* production as lighting represents one of the highest operating costs and associated environmental performance [22]. The impact of the light spectrum on *C. sativa* cultivation has been reported with different lighting systems, including HPS [23,24] and LEDs [25,26,27]. These studies concluded that blue light led to increased cannabinoid concentration (% *w/w*), while supplemental green light-induced both cannabinoid and terpene accumulation [17,27]. Blue light beneficially affects chloroplast development and stomatal opening [28,29,30]. Most commercial lighting fixtures designed for *C. sativa* production have a high percentage of blue light. However, some studies have shown that an increase in blue-light fraction causes yield reduction in leafy greens [31,32] and in *C. sativa* inflorescence dry mass and cannabinoid yield [17,33]. Increasing the blue light-fraction for *C. sativa* inflorescence production does seem to compromise the overall cannabinoid production profitability [34].

Light spectrum greatly influences terpene and cannabinoid accumulation, as both share the same primary biosynthesis step [16,17,35]. The profile of this photo-protectant organic molecule impacts the flavor quality of *C. sativa* inflorescence, as well as impacting its therapeutic effect, highlighting the importance of terpene accumulation in *C. sativa* production. However, limited information on the impact of the light spectrum on terpene diversity can be found. The majority of *C. sativa* research to date has primarily focused on the impact of light spectrum on inflorescence yield and cannabinoid accumulation, neglecting terpene profiling and quantification [26,33,34,36] 

It is difficult to dissect the impact of individual wavelengths on *C. sativa* morphological traits and secondary metabolite production with these varied spectra. It makes comparison among studies difficult since it is unknown if such beneficial responses were triggered by a potential synergistic impact of combined wavelengths. This ultimately proves challenging when determining what light wavelengths are essential for growth traits and secondary metabolite production in *C. sativa*.

Studies have reported that the use of blue light LEDs can improve terpene profiles in herbs such as *Thymus* species [37], *Perovskia* species [38], Japanese mint [39], and others [40]; yet this has not been reported for *C. sativa*. Furthermore, available *C. sativa* studies are rarely conducted under a monochromatic spectrum. Fixtures providing a mix of blue and red light or the use of LEDs as supplemental lighting are often reported in the literature [17,25]. It was hypothesized that monochromatic blue light could be used to increase secondary metabolite production in *C. sativa.* As for red light, it has been associated with stem growth and rooting in other plants [2]. This seemed to translate to better cutting development in *C. sativa* exposed to high-red content lighting [18]. However, no studies have looked at the influence of monochromatic red or blue light on secondary metabolite production and biomass development in *C. sativa*.

The impact of amber light (595 nm) on *C. sativa* production has not been investigated. Its use as horticultural lighting is promising because its spectrum is similar to that of HPS lamps while being much more energy efficient [41]. Superior yield was achieved in certain cases when compared to dichromatic B:R LED lights in other plants [42,43].

The objective of this study was to investigate and evaluate the impact of light spectra, including monochromatic light (blue, and red narrow wavelengths, in addition to a wide-amber wavelength) and mixed light spectra with different B:R ratios than ones used in previous experiments, to pinpoint an optimal B:R ratio [26,27]. Intermediate chemotype *C. sativa* plants with both THC and CBD were cultivated under six light treatments, including five different LED spectra and an HPS spectrum in a controlled environment. *C. sativa* inflorescence yield, energy efficiency, secondary metabolite (cannabinoid and terpene) yield, and profile were compared between each light treatment.

Results from this study provide detailed information on how the light spectrum impacts *C. sativa* production and accumulation of secondary metabolites. We hypothesize that different percentages of blue light can impact inflorescence yield, which negatively deteriorates the overall secondary metabolite yield (i.e., g plant^−1^). As LEDs are gaining momentum as a standard lighting system for production facilities in this nascent legal industry, these data may help when constructing an optimal spectrum for *C. sativa* growth.

## 2. Results

*C. sativa* ‘Babbas Erkle Cookies’ plants were cultivated under six different light spectra. Each light fixture had its light spectra confirmed by spectroradiometer analysis (Figure 1).

### 2.1. Morphology

#### 2.1.1. Plant Height and Vertical Growth Rate

Seedling height at the start of replicate 1 was significantly higher (*p* < 0.0001) than seedling height at the start of replicate 2: 15.5 ± 2.2 cm versus 8.2 ± 1.1 cm respectively. The final height was significantly higher (*p* < 0.0001) for replicate 1 versus replicate 2: 76.1 ± 7.3 cm and 70.4 ± 4.9 cm respectively (Appendix A). Table 1 summarizes plant height at harvest per light treatment per replicate. Plants cultivated under amber light resulted in the tallest plants for both replicates, while blue light resulted in the smallest plants for both replicates. A varying effect between replicates was observed for the purple, rose, amber, red, and HPS light treatments (Table 1).

Growth rate had no significant difference between replicates. Light treatment significantly affected the growth rate in the vegetative phase and in the first three weeks of the flowering phase (*p* < 0.0001). During the vegetative phase, the highest growth rate was achieved with red light, while the lowest growth rate was achieved with purple, blue, or HPS. For the flowering phase, amber light had the highest growth rate while blue had the smallest recorded growth rate (Figure 2).

#### 2.1.2. Tissue Coloration

Leaf and inflorescence coloration for *C. sativa* plants cultivated under different light treatments were compared one week before harvest (Figure 3). Color histogram results (RGB values) showed that different lights did not significantly influence inflorescence coloration. Rather, leaf coloration was affected and similar RGB profiles were observed for plants cultivated under HPS and Rose lights. A higher fraction of green color was observed for plants cultivated under Amber and Red lights, and leaves grown under these two light treatments were visibly brighter green than plants cultivated under the other light treatments. Both Purple and Blue lights resulted in lower RGB values, and Blue light resulted in the lowest fraction of blue color.

### 2.2. Inflorescence Yield

Significant differences (*p* < 0.05) were observed between replicates for average fresh inflorescence yield with 116.2 g plant^−1^ ± 16.6 and 104.5 g plant^−1^ ±18.3 for replicate 1 and 2 respectively. Similarly, mean dry inflorescence was significantly different (*p* < 0.0005) between replicate 1 and 2, with average values of 23.3 g plant^−1^ ± 3.7 and 19.4 g plant^−1^ ± 3.7 (*p* = 0.0004) for replicate 1 and 2, respectively (Appendix A). The mean inflorescence fresh and dry mass for each light treatment per replicate is summarized in Table 2. HPS resulted in the highest average fresh and dry inflorescence mass for both replicates. Light intensity was significantly different between replicates (*p* < 0.0001), with replicate 1 having ~7% more PPFD per plant. Light treatments had a slightly different effect when ranking inflorescence yield between replicates for light treatments rose, purple, amber, and red. Blue LED light resulted in the lowest average fresh and dry inflorescence mass in both replicates. Drying reduced inflorescence mass by approximately 80% for each light treatment, and the moisture concentration was between 12.5 ± 1.5%.

### 2.3. Phytochemical Concentration

Cannabis plant inflorescence was analyzed to determine the concentration of the major cannabinoids (THC, CBD, and CBG) and terpenes. Significant differences (*p* < 0.005) were observed for THC and CBD concentrations between replicates, but not for CBG and terpenes (Appendix A). Table 3 summarizes the effect of light treatments on total THC, CBD, and CBG concentration per replicate.

#### 2.3.1. Total Cannabinoid Concentration

A significant difference in total THC and CBD concentration was observed between replicates (*p* < 0.005; Appendix A). Higher total THC concentration was observed in the second replicate versus the first. The highest total THC concentration was observed in the blue light treatment in both replicates (Table 3). Purple ranked second in total THC concentration for both replicates. Light treatment HPS, amber, red, and rose had a variable effect on total THC concentration between replicates. The lowest total THC concentration was observed in the amber and red treatment.

Differences in CBG concentration between replicates were not statistically significant (Appendix A). A significant impact of light treatment on total CBG concentration was shown (*p* < 0.0001), with blue and purple light having the highest average concentration with 0.18 ± 0.03 % and 0.15 ± 0.03% total CBG respectively. Amber and Red showed the lowest concentration with average values of 0.05 ± 0.02% total CBG for both replicates (Table 3).

#### 2.3.2. Total Terpene Concentration

Total terpene concentration was not significantly different between replicates (Appendix A). Terpene production significantly varied depending on light treatment (*p* < 0.0001). Purple light treatment resulted in the highest concentration of terpenes produced (29.43 mg g^−1^ ± 1.15), followed by Blue (28.48 mg g^−1^ ± 1.14), HPS (25.83 mg g^−1^ ± 1.00), rose (24.51 mg g^−1^ ± 1.38), red (20.30 mg g^−1^ ± 2.16), and amber (18.10 mg g^−1^ ± 1.30) light treatments.

#### 2.3.3. Light Effect on Terpene Profile

A total of 87 terpenes compounds were quantified (Appendix A), and the six most abundant (500 g mg^−1^ in at least one treatment) are presented in Table 4. Myrcene was produced more than any other terpene for all light treatments. Purple light produced the most myrcene (8.91 ± 0.53 mg g^−1^), followed by blue, HPS, rose, red, and amber light. Amber was significantly less than purple (*p* ≤ 0.001), blue (*p* = 0.008), rose (*p* = 0.042) and HPS (*p* = 0.027). α-Pinene was the second most abundant produced terpene for inflorescence cultivated under all light treatments. Blue light produced the most α-pinene (5.73 ± 0.38 mg g^−1^), followed by purple, HPS, rose, red, and amber light. Limonene was the third overall most produced terpene and inflorescence cultivated under purple light produced the most limonene (3.05 ± 0.12 mg g^−1^), followed by HPS, blue, rose, red, and amber light. Amber has significantly less limonene than purple (*p* ≤ 0.001), HPS (*p* = 0.041) and blue (*p* = 0.044). Red has significantly less limonene than purple (*p* = 0.023). Purple light produced the most β-pinene (2.70 ± 0.12 mg g^−1^), followed by blue, HPS, rose, red and amber light treatments. Purple and blue were significantly greater in β-pinene than amber (both *p* = 0.003) and red (Purple *p* = 0.011, Blue *p* = 0.014). Inflorescence cultivated under blue light produced more linalool than other light treatments (0.847 ± 0.04 mg g^−1^), followed by purple, HPS, rose, red, and amber light. Amber was significantly less than purple and blue (both *p* ≤ 0.001). (E)-β-ocimene was produced mostly in the purple (0.75 ± 0.07 mg g^−1^), followed by blue, rose, HPS, red, and amber light. Amber has significantly less (E)-β-ocimene than purple (*p* ≤ 0.001), blue (*p* = 0.022), and rose (*p* = 0.026). Purple had significantly greater (E)-β-ocimene than red (*p* = 0.009).

### 2.4. Total Phytochemical Yield

#### 2.4.1. Light Treatment Impacts Total THC and CBD Yield per Plant

Light treatment had a significant impact on THC and CBD yield per plant (*p* < 0.0004 and *p* < 0.0001 respectively). Total THC yield was highest when plants were cultivated under HPS (Figure 4). The highest total THC yield per plant was found in HPS (2.54 ± 0.29 g plant^−1^), followed by rose (1.98 ± 0.16 g plant^−1^), purple (1.81 ± 0.12 g plant^−1^), red (1.60 ± 0.24 g plant^−1^), blue (1.44 ± 0.11 g plant^−1^) and amber (1.18 ± 0.15 g plant^−1^) light. For total CBD yield, the highest was found in HPS (1.76 ± 0.19 g plant^−1^), purple (1.16 ± 0.08 g plant^−1^) rose (1.4 ± 0.09 g plant^−1^), red (1.1 ± 0.17 g plant^−1^), blue (0.86 ± 0.07 g plant^−1^) and amber (0.84 ± 0.11 g plant^−1^).

#### 2.4.2. Light Treatment Impacts Total THC and CBD Yield per Plant

Light treatment had a significant impact on CBG and terpene yield per plant (*p* < 0.0001 and *p* < 0.01, respectively; Figure 5). CBG yield was highest in purple (0.029 ± 0.003 g plant^−1^), HPS (0.027 ± 0.004 g plant^−1^), blue (0.024 ± 0.002 g plant^−1^), rose (0.021 ± 0.003 g plant^−1^), red (0.012 ± 0.003 g plant^−1^) and amber (0.009 ± 0.002 g plant^−1^). Total terpenes produced per plant were greatest in plants cultivated under HPS light (0.78 ± 0.08 g plant^−1^). Purple (0.60 ± 0.06 g plant^−1^) and rose (0.60 g ± 0.05 plant^−1^) produced equivalent amounts of terpenes, followed by red (0.49 ± 0.07 g plant^−1^), blue (0.41± 0.05 g plant^−1^) and amber light treatments (0.34 ± 0.04 g plant^−1^).

### 2.5. Light-Associated Cannabinoid Production Efficiency

Light-associated cannabinoid production efficiency (CPE_light_) per plant showed significant differences for each light treatment. HPS and Rose had the lowest energy consumption per g of cannabinoid produced with ~290 kWh g^−1^ THC and ~410 kWh g^−1^ CBD. The highest energy consumption by a unit of cannabinoid was seen with the amber light treatment (Figure 6).

## 3. Discussion

This work explored the effects of six different light spectra on *C. sativa* morphology, yield, and light-associated production costs. We investigated how changes in light spectrum influence plant morphology and key secondary metabolite production: cannabinoids and terpenes. Light spectrum composition highly impacted growth traits and secondary metabolite profiles, under the same light intensity, which had a significant impact on lighting associated CPE_light_.

### 3.1. Morphology

Light treatment with a high blue fraction caused smaller plant height and vertical growth rate, as has been observed in other studies [26,27]. A high fraction of blue light, or blue-dominant light, resulted in the lowest plant height. This was expected, as dwarfing of plants grown under blue-dominant light has been observed for other greenhouse crops [44,45]. Blue light induces cryptochrome photoreceptor activation, which slows vertical plants, although not all plants experience shortening with blue light [46]. This typical blue-light-induced dwarfism in *C. sativa* plants under blue is consistent with the observation in other plants, where cryptochrome seems to be involved.

Amber light resulted in the greatest plant height while monochromatic red light resulted in smaller plants and a significantly lower growth rate during the flowering phase. In other plants, the stimulation of internodal stretching and subsequent plant elongation by red-light induced shade avoidance response through phytochrome activation [47]. Red-light-induced vertical growth increases have been previously reported in *C. sativa* [18] and plant elongation of other greenhouse crops [48,49,50]. LEDs with a higher proportion of red wavelength increased plant height in other studies [26,33], but no study investigated the effect of amber light on cannabis plants.

The molecular mechanism of amber light-induced vertical growth in *C. sativa* during flowering is unknown. It could be hypothesized that this occurs through different molecular pathways other than the characterized shade avoidance response/photoreceptor activation, as amber light is not believed to activate phytochrome [41]. The higher plant height observed with the amber light treatment could be caused by specific protein over-expression. Proteomic analysis on *Arabidopsis thaliana* showed that amber light results in a 2- to 6-fold increase in the expression of key proteins implicated in carbon and amino acid metabolism (GAD2, RBCS-1A), protein synthesis, folding, and degradation (HSC70–1), as well as cytoskeleton and cell wall (PME3) [51]. Increased expression of these proteins could cause the observed internodal stretching.

Regardless of the mechanism implied in amber-light-induced vertical growth, growers want better control of plant height. Whether to achieve better air circulation in the canopy or homogenize canopy height, amber light could be used instead of red light. Integrated plant management programs focusing on promoting canopy air flow to decrease fungal infection in the phyllosphere might consider incorporating additional amber lighting to the grow room. Known greenhouse pathogens such as *Botrytis*, *Fusarium*, and *Phytium* are affected by humidity management via airflow increase [52,53]. Conversely, growers wanting to increase productivity via vertical farming could benefit from blue light-induced smaller plants. Vertical farming increases facility productivity for some crops but studies on cannabis are lacking [54].

### 3.2. Inflorescence Mass

Inflorescence mass is a key factor in determining cannabis yield for growers. HPS had the greatest inflorescence mass, followed closely by Rose and Red light. A study conducted by Magagnini et al. [18] supports these results, reporting that *C. sativa* plants grown under HPS yielded more inflorescence than plants grown under LEDs. Although some similarities in the spectral compositions of HPS and amber light can be noted, the latter yielded 30% less fresh inflorescence mass. Differences between these two spectra had HPS light with a lower percentage of 595-nm light, higher peaks from 420 nm to 460 nm, and 500 nm, when compared to the Amber LED spectrum. These differences in spectral composition, particularly for 500-nm light, may be why there is a considerable difference in inflorescence mass between HPS and Amber. However, when comparing Red and Rose light treatments, supplementing with a small fraction of blue light does not seem to impact fresh inflorescence mass.

Danziger and Bernstein [26] tested different light spectra of three different chemotypes. Their study used LED with a B:R ratio of 1:1 and 1:4 versus the B:R ratios used in the present study (2:1 and 1:10). Inflorescence yield was almost doubled compared to HPS when the intermediate chemotype ‘CS12′ was grown under a LED light having a 1:4 B:R ratio. It should be mentioned that only one of the three chemotypes tested in their study had a significantly higher yield under certain LEDs when compared to HPS. Of note, the light intensity used in this study is half of what was used in their study (950 versus 400 μmol m^−2^ s^−1^). Other studies, such as the one performed by Wei et al. [33] did not normalize PPFD between treatments. The question of a potential interaction between light intensity and spectrum composition should be part of future studies. If growers want to achieve higher yield with LEDs, they should consider the need for fine-tuning the B:R ratio, light intensity, and chemotype selection.

### 3.3. Cannabinoids Concentration, Yield, and Profitability

In this work, plants grown under blue-dominant light (blue or purple) yielded the highest THC and CBG concentration but it did result in lower inflorescence mass when compared to the control. This study aligns with others indicating that THC concentration increases with increasing blue light [18,26,27]. It should be highlighted that whereas the Rose spectrum (B:R ratio 1:10) and HPS had similar inflorescence mass, the difference in THC percentage did cause a significant decrease in g THC plant^−1^. At a glance, the difference between the two spectra is a 6-fold increase in light intensity in the 700–799 nm range for HPS, as well as an absence of any wavelength >800 nm in the Rose LED spectrum. Influence of the Far-Red spectrum (700–780 nm) on cannabinoid yield should be further investigated.

Unlike THC, CBD concentration did not seem to respond as much to increasing fractions of blue light. Comparable CBD percentages were found in the inflorescence of plants cultivated among blue, purple, rose, and HPS light treatments, while lower CBD percentages were observed in amber and red treatment.

Conflicting results have been reported regarding the effect of blue light on CBD biosynthesis depending on chemotypes. THC-dominant chemotype (drug-type) *C. sativa* seems to increase CBD biosynthesis with increasing blue light fraction [17,18,27]). Increasing the blue-light fraction has conflicting results in CBD-dominant (fiber-type) chemotypes. Some studies report no significant impact of light spectrum on CBD biosynthesis [17,34], whereas others did report a significant change in CBD concentration for plants grown under a higher blue-light fraction [18,25,33]. Studies reporting increased CBD concentration with an increase in blue-light fractions did compare treatment with different PPFD levels. The increase in CBD concentration could be solely caused by an increase in PPFD. Interestingly, light treatment in other studies impacted cannabinoid biosynthesis so much that it blurred previously described legal thresholds. For example, Westmoreland et al. [34] showed how legally defined hemp (THC < 0.03%) is transformed into legally defined drug-type cannabis (THC > 0.03%) [55,56] if grown under higher blue-light fractions. Further studies should consider chemotype-specificity when measuring light quality-driven cannabinoid biosynthesis, as CBD-dominant chemotypes seem to be less receptive to blue-light-driven increases in cannabinoid biosynthesis.

There was a consistent increase in CBG concentration as the fraction of blue light increased between the Rose, Purple, and Blue treatments. This finding agrees with other reports where LED light containing blue light resulted in higher CBG concentration than HPS [18]. As CBG is still being studied for new medical indications [57], strategies to increase its concentration are worth investigating further. Again, the interaction between light quality and chemotypes should be studied further, especially for growers working with CBG-dominant (type IV) chemotypes [58].

Cannabinoid concentration in *C. sativa* should always be analyzed in conjunction with biomass accumulation. In this study, HPS treatment produced the most cannabinoids (THC and CBD) on a per-plant basis. The light treatments containing blue light (blue and purple) resulted in 22 to 43 % less total THC yield per plant. It is important to note that although blue light may induce higher THC and CBD percentages in inflorescence biomass, it appears to suppress inflorescence growth, resulting in less total secondary metabolite produced per plant. This could be of concern to growers as blue light can lead to a greater reduction in overall cannabinoid yield.

Of further industrial relevance, we suggest reporting cannabinoid yield as a per-plant metric. This metric is relevant from a grower’s perspective, since it shows how much THC was produced per plant per growing cycle, rather than presenting THC concentration percentage-wise (% mass or % m/m) [2,18]. These units should facilitate the translation of scientific findings to growers using different plant densities in their facilities.

Using a novel CPE_light_ metric, it was possible to show how LEDs can help produce cannabinoids more efficiently. The same amount of energy was necessary to produce 1 g of THC under HPS than Rose or Purple, even when plants had significantly lower inflorescence mass or THC concentration. Because lighting is one of the main operational costs for growers, increasing light intensity in an already operational grow room could be rendered more economical if using LED lamps instead of adding extra HPS lamps.

### 3.4. Terpenes

Terpenes are responsible for inflorescence odor and flavor profiles [59]. Light treatments with blue-dominant spectra led to higher total terpene production, monoterpenes, and sesquiterpenes. Purple light treatment resulted in the highest total terpene concentrations (mg g^−1^), followed by the blue light treatment. Higher concentrations of monoterpenes and sesquiterpenes were observed under these two light treatments, with a higher fraction of blue light, and this data agree with a previous study [17], in which higher concentrations of monoterpenes such as α-pinene and limonene were reported with supplemental blue, green and red light. Although there was a relatively low fraction of blue light in the HPS treatment, it induced comparable total terpene concentrations to the Blue light treatment. Although different cultivation approaches were applied, this finding regarding HPS treatment agrees with that of Namdar et al. [27]. The authors reported that a higher total terpene concentration was observed when plants were grown under LEDs with a high B:R ratio in the vegetative phase and flowered under HPS light. In this work, the same spectral treatment was applied throughout the vegetative and flowering stages, and higher total terpene concentrations were observed under both HPS and purple treatments.

### 3.5. Limitations (Differences between Replicates)

Significant differences in inflorescence mass (fresh and dry), total THC%, and total CBD% were observed between two temporal replicates, and plants with different origins may have affected how the plants grew and developed. The plants used to conduct replicate 1 were transported over a two-day period and arriving in a closed box, suggesting there was little air exchange for two days. Seedlings from replicate 1 were on average double the height as the ones used for replicate 2. Plants used for replicate 2 were cloned from the original lot of plants received, as the original licensed producer stopped shipping seedlings mid-way through replicate 1. Plants were given a week of standardized lighting before being placed in a treatment to get their growth rate as similar as possible. Seedling plant height might be a cause for the observed significant difference in final harvested inflorescence and phytochemical concentration.

Other environmental parameters that exhibited differences between replicates were PPFD levels (5–10% difference) and night-time temperatures (4 °C). The smaller seedlings from replicate 2 were exposed to colder nights and lower PPFD. They produced less inflorescence but were more concentrated in phytochemicals. Interestingly, higher PPFD in the first replicate should have resulted in higher cannabinoid concentration, as observed in other studies [23,60]. Future studies could look at the effect of seedling height and night-time temperature on phytochemical biosynthesis.

### 3.6. Implication for Growers

Understanding that the light spectrum considerably affects growth and secondary metabolite derivation in *C. sativa* is of utmost importance for growers wanting to sustainably increase yield in their indoor facilities. Monochromatic blue lights exemplify how secondary metabolite production can be manipulated, yet this may prove disadvantageous for growth traits such as inflorescence size. One possible application of this could be intermittent blue light supplementation, especially at the end of the growth cycle, which could cause an increase in THC without greatly affecting biomass.

Dichromatic LED light is a possibility for balancing out the deleterious effects of pure blue light on growth traits while maintaining secondary metabolite levels that are comparable to conventional HPS light. Inflorescence yield is linked to the HPS spectrum, which is amber-rich light containing a low fraction of blue light. When considering whole plant yield to quantify cannabinoid levels of a given crop (g plant^−1^), HPS light resulted in the highest cannabinoid concentration, whereas LED light with different blue-red light ratios lowered cannabinoid yield.

It is important for growers to consider the impact of individual light wavelengths on both growth traits and secondary metabolite production since inappropriate spectral design could lead to a greater reduction in overall THC yield. The fact that the CPE_light_ of rose and purple are similar to HPS might be a part of the solution for growers who want to increase their light intensity without increasing their operational cost. This study highlights the importance of optimizing spectral conditions for maximizing cannabis production and decreasing operational costs. Future studies could expand on this research by deploying light qualities richer in the amber region of the spectrum and modifying the fraction of blue light. Hybrid ‘grow rooms’, having both HPS and LEDs might be a more profitable way to increase light intensity.

## 4. Materials and Methods

### 4.1. Cultivation Environment

The cultivation room comprised two 2.4 m × 1.2 m and two 1.8 m × 1.2 m flood tables, creating six separate 1.2 m × 1.2 m treatment areas in an indoor environment (Montréal, Quebec, Canada). Each treatment area was covered by 80% black shade cloths that were doubled layered to reduce stray light (~96% reduction). Shade curtains were used on three walls with the front side open. For each flood table, a 50-cm void space was left at the base under each shade curtain to allow air circulation. An exhaust fan cycled air between the growth room and the outside environment, and four fans were placed in the room to allow for constant air movement. For each light treatment, temperature and humidity were recorded hourly throughout the experiment with Hobo sensors (S-THB-M002, OnSet, Hobo, Bourne, MA, USA). During the day, the temperature of the grow room was approximately 28 ± 2 °C, and relative humidity was between 40–55%. Night-time temperature was 25–27 °C for Replicate 1 and 19–21 °C for Replicate 2; both replicates maintained a night-time relative humidity between 50–65%. In each replicate, the differences in temperature and relative humidity among the treatments (light treatment zones) were less than 2% and 4% respectively.

Plant density and spacing between plant pots in each treatment area were adjusted based on plant growth stages. At vegetative stages, each treatment area (1.2 m × 1.2 m) contained 12 plant pots, oriented in three rows with 0.25 cm spacing in between (8.3 plants m^−2^). At the beginning of the flowering stage, three plant pots were removed to ensure proper spacing between plants (6.25 plant m^−2^). Trellising nets were installed horizontally 15 cm over the plants to serve as support. Strings were added to support and maintain the vertical orientation of the plants. Above each treatment area, one light was suspended above the center of each treatment area, with a system of pulleys and cables. This allowed a consistent light intensity among the treatments via height adjustment of the light fixtures.

### 4.2. Light Treatments

The six light treatments used in this experiment were as follows: (1) HPS light (control; Gavita Pro 6/750e DE FLEX, Aalsmeer, Netherlands), (2) ‘Amber’ light; 595 nm monochromatic wide spectrum (VQ-GLIB600W-595, Vanq Technology, Shenzhen, China); (3) ‘Red’ light; 630 nm monochromatic narrow spectrum (VQ-GLIB600W-630, Vanq Technology, Shenzhen, China); (4) ‘Rose’ light; 430 nm and 630 nm polychromatic narrow spectrum with a 1:10 B:R ratio (VQ-GLIB600W-630/430n, Vanq Technology, Shenzhen, China); (5) ‘Purple’ light; 430 nm and 630 nm polychromatic narrow spectrum with a 2:1 B:R ratio (VQ-GLIB600W-630/430m, Vanq Technology, Shenzhen, China) and (6) ‘Blue’ light; 430 nm monochromatic narrow spectrum (VQ-GLIB600W-430, Vanq Technology, Shenzhen, China). All LED spectra emitted 600 W from four 150 W LED chips equipped with glass circular optic lenses (90° viewing angle). The HPS lamp was a larger single-cylinder bulb with an aluminum refection hood, which was powered at 750 W. All spectral composition data were confirmed with a spectroradiometer (ALP00051300010731, Asensetek, Gatineau, QC, Canada). Electrical parameters were measured with a current monitor (KW47-US, Kuman Tech, Shenzhen, China). The lighting fixture and spectra are listed in Table 5.

The average (photosynthetic photon flux density) PPFD level and photoperiod were adjusted based on plant growth stages. During the first two weeks (days 0–13, vegetation stage), plants were grown with an 18 h d^−1^ photoperiod at 250–270 μmol m^−2^ s^−1^. At the end of the vegetative stage (day 13), the remaining nine plants were switched to inductive photoperiod (12 h d^−1^) for 8 weeks (day 14–70) until harvest. The light uniformity in each light treatment was between 0.91 to 0.94; this was determined by the ratio of the minimum to average PPFD level as described by Balasus et al. [61]. During cultivation, plants were randomly reorganized every 3 days, which avoided inconsistent PPFD levels caused by light uniformity. In the meantime, PPFDs under each light treatment were confirmed and adjusted to the set point by adjusting the height of the lights. Once being switched to inductive photoperiod, PPFD levels were increased by ~20 μmol m^−2^ s^−1^ every week until plants stopped growing vertically (day 35, third week of flowering), then maintained at 400 μmol m^−2^ s^−1^. Stray light testing was performed with all lights on, except the treatment zone being tested. Measurements were made at the center base of the table and stray light bleeding was negligible (<2%).

### 4.3. Plant Materials and Cultivation

One hundred cuttings (100) of *C. sativa* from the ‘Babbas Erkle Cookies’ accession (WeedMD Inc., Aylmer, ON, Canada) were obtained. This accession is an intermediate chemotype (Type II; THC concentration> 0.3% and CBD concentration >0.5%) [62]. Out of the initial 100 cuttings, 72 plants uniform in size, 12 per experimental group, were transplanted and placed in a growth chamber for the first crop cycle (Replicate 1). The remaining plants were used as mother plants. Cuttings for the second crop cycle (Replicate 2) came from these mother plants. Mother plants were maintained under fluorescent light (RAZR2, Fluence, Austin, TX, USA). Cuttings for Replicate 2 were rooted using indole-3-butyric acid gel (Technaflora, Mission, BC, Canada) and rapid rooter plugs (General Hydroponics, Santa Rosa, CA, USA), with an 18 h d-1 photoperiod and a PPFD level of 125 μmol m^−2^ s^−1^ under high humidity (>90%), using a propagating tray with transparent dome cover. PPFD was determined with an LI-250A Light Meter and an LI-193 Spherical Underwater Quantum Sensor (LI-COR, Lincoln, NE, USA). Successfully propagated cuttings showing adventitious roots were transplanted into 750 mL square pots with Canna-coco coconut husk mixture (Canna, Toronto, ON, Canada), and 10 mL Myke Tree and Shrub Mycorrhizae were added to the pots (Premier Tech, Rivière-du-Loup, QC, Canada).

Plants were irrigated with the commercial nutrient solution every 2–4 days. Nutrient solutions comprised tap water (Montreal, QC, Canada), Coco A&B nutrient solutions (Canna, Toronto, ON, Canada), KH_2_PO_4,_ and K_2_SO_4_. Electrical conductivity (EC) and pH were monitored with a handheld EC and pH meter (HI98130, Hanna Instruments, Woonsocket, RI, USA) and pH was corrected to 6 with an H_3_PO_4_ solution (HGDi Technologies, Montreal, QC, Canada) before irrigation. Representative samples were analyzed by an independent laboratory (A&L Canada, London, ON, Canada). The average electrical conductivity (mean ± SE, *n* = 25) was 1.9 ± 0.3 mS cm^−1^ with nutrient element concentration of N 137.1 ± 6.4 mg L^−1^, P 88.9 ± 15.0 mg L^−1^, K 149.9 ± 20.2 mg L^−1^, Ca 141.7 ± 5.9 mg L^−1^, Mg 44.7 ± 1.8 mg L^−1^, S 86.4 ± 16.7 mg L^−1^, Cl 10.5 ± 0.3 mg L^−1^, Fe 0.6 ± 0.02 mg L^−1^, Zn 0.2 ± 0.01 mg L^−1^, Mn 0.3 ± 0.01 mg L^−1^, B 0.2 ± 0.01 mg L^−1^, and Cu 0.047 ± 0.001 mg L^−1^.

Total nitrogen was analyzed by combustion/thermal conductivity (Dumas method), and chloride by K_2_SO_4_ extraction with standard method 4500-Cl- G Mercuric Thiocyanate Flow Injection Analysis. Other elements were analyzed by acid digestion (ICP-OES Ref. EPA3050B/EPA6010B).

The first watering was performed by hand to compact the coco grow medium and sequential watering was provided by two submersible pumps (728305, EcoPlus, Austin, TX, USA) every 2–4 days. The amount of water provided increased during the study and tripled once plants were transplanted into their final pots. No nutrients were added to the irrigation water from week 8 until harvest at week 10. Plants were trimmed on day 35 (third week of flowering) using the following guidelines: (1) All fan leaves on the bottom two-thirds of the primary stem were removed; (2) branches that grew from the bottom one-third of the main stem were stripped of all fan leaves except for the top two fan leaves while all inflorescence leaves were kept; (3) Leaves in contact with inflorescences were removed. This technique is similar to the bottom branches and leaves removal (BBLR) method described by Danziger et al. [26].

### 4.4. Vertical Growth Rate

Plant vertical growth rate was recorded under each light treatment. Height was measured weekly from the base of the plant to its highest point (excluding leaves). The vertical growth rate for the flowering period was measured only for the first three weeks which is when most (>90%) of the vertical growth happens [26]. The growth rate was calculated with Equation (1):(1)$$\Delta H=H_{n+1}-H_{n}$$
where ΔH is vertical growth rate, $H_{n}$ is plant height at the start of a given week (cm) and $H_{n+1}$ is plant height at the start of the following week (cm).

### 4.5. Plant Images

Plant images were taken one week before harvest. For this purpose, the lighting from each treatment was switched off and a flash from a digital camera was used. The camera was placed above the plant canopy, and the distance from the camera to the plant canopy was consistent (46 cm). Images were analyzed with a color histogram using ImageJ 1.48v software (Bethesda, MD, USA), to determine the effect of each light treatment on inflorescence and leaf coloration.

### 4.6. Plant Harvest and Phytochemical Measurement

Plants were harvested eight weeks after initiating the flowering photoperiod. Inflorescence was weighed separately and placed on a screen for drying. The inflorescence was placed in a dark room with a dehumidifier at 18 °C, to a moisture concentration of 12.5 ± 1.5%. Dried inflorescence was placed into a plastic bag for curing. Over a one-week curing period, bags were opened once a day for 10 min.

Cannabinoid and terpene analyses were performed on dried plant material from three plants in each treatment. Three separate 4-g inflorescence samples per plant were subjected to cannabinoid and terpene analyses, and each 4-g sample comprised a portion of the biggest inflorescence as well as equal parts of medium and small-sized inflorescences. Cannabinoid and terpene analyses were performed by Laboratoire PhytoChemia (Saguenay, QC, Canada).

Measured cannabinoids included THC, tetrahydrocannabinolic acid (THCA), CBD, cannabidiolic acid (CBDA), cannabigerol (CBG) and cannabigerolic acid (CBGA). Cannabinoids were analyzed based on the procedures described by De Backer [43]. Briefly, 0.25 g samples were mixed with extraction solvent (methanol:dichloromethan, 9:1). The sample was filtrated before injection in a high-performance liquid chromatograph (HPLC). The column used was Kinetex 2.5 μM C18, 150 × 4.6 mm, and the detector was a diode-array (DAD). The primary mobile phase was deionized water with formic acid (0.1%) and the secondary mobile phase was methanol:acetonitrile (75:25) with formic acid (0.1%). References used for THCA, CBDA, CBGA, THC, CBD and CBG have CAS numbers 23987-85-0; 1244-58-2; 25555-57-1;1972-08-3; 13956-29-1; 25654-31-3 respectively. Methyl octanoate at ≥99% purity has a CAS of 111-11-5.

Terpenes were analyzed by mixing 1 g of dried biomass with 5 mL of extraction solvent: pentane with octanoate methyl (0.16 mg L^−1^). Samples were injected in Agilent 127-5012 DB-5 and 127-7012 DB-WAX columns for gas chromatography and mass spectrometry (GC-MS). Identification was made by spectral matching with available databases [63,64,65,66,67]. Instrument parameter and quantification of unique terpenes was based on guidelines by the International Organization of the Flavor Industry (IOFI) [68].

Total THC was defined as Equation (2). This equation was applied to total CBD. For CBG the value used was acidic form × 0.878
(2)Total cannabinoid content=Acidic form×0.877 or 0.878+Neutral form
where total cannabinoid concentration, acidic form and neutral form is in percent (%). Phytochemical yield per plant was calculated following Equation (3).
(3)Total phytochemical yield=Total phytochemical content×Dry inflorescence mass
where total phytochemical yield is in g plant^−1^, total phytochemical is in % or mg g^−1^ and dry inflorescence mass is in g. No significant difference in average phytochemical yield per plant between replicates was observed (Appendix A).

### 4.7. Light Associated Cannabinoid Production Efficacy

Each light treatment had its cannabinoid production efficacy (CPE_light_) measured for THC and CBD. Energy used by the lamp was divided per quantity of cannabinoid produced per plant and was calculated following Equation (4).
(4)$$CPE_{light}=P_{light}\times Operation\;time÷Total\;cannabinoid\;yield$$
where CPE_light_ is the cannabinoid production efficacy (kWh g^−1^), P_light_ is lamp power (kW) with values of 0.6 kW for LED lamps and 0.75 kW for HPS, operation time is in hours (h) with a value of 924 and total cannabinoid yield in g cannabinoid.

### 4.8. Statistical Analysis

The study was a randomized complete block design with replicates as blocks. To eliminate a potential grow room area effect, treatments were randomly assigned to an area in the grow room between replicates. Each plant was treated as an experimental unit. For morphological analysis (height, weight) the sample size was nine per treatment. For phytochemical analysis, a subgroup of three homogeneous plants were chosen. Statistical analyses were performed using JMP 14.1.0 software (SAS Institute, USA). The difference between replicates was assessed by student T-test using Kenward and Roger correction for degrees of freedom. This method provides more accurate tests for smaller samples. If no significant difference between replicates was observed, results from two replicates were pooled and submitted to a one-way ANOVA with Tukey HSD posthoc tests and light treatment as the fixed effect. Only descriptive statistics are used if results between replicates were significantly different.

## Figures and Tables

**Figure 1 plants-11-02982-f001:**
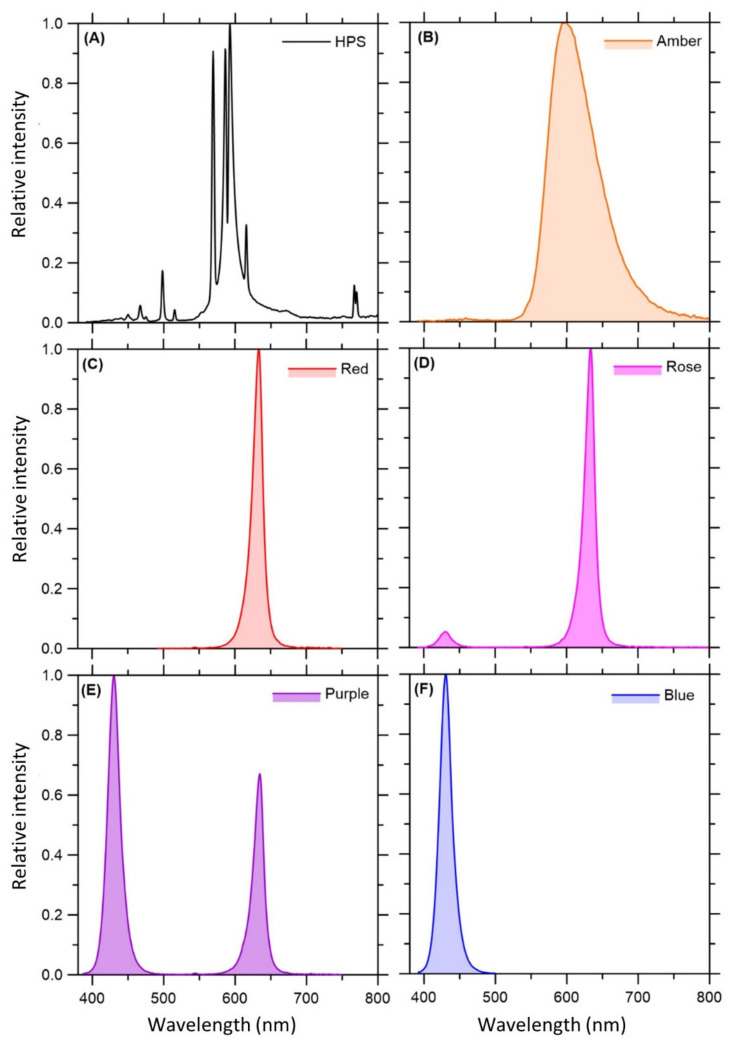
Relative spectra of an HPS lamp (control) and five experimental LED light treatments for cannabis plant cultivation. (**A**) HPS, (**B**) Amber (wide spectrum 595 nm), (**C**) Red (narrow spectrum 630 nm), (**D**) Rose (narrow spectrum 430 nm and 630 nm with a B:R ratio of 1:10), (**E**) Purple (narrow spectrum 430 and 630 nm spectrum with a B:R ratio of 2:1), and (**F**) Blue (narrow spectrum 430 nm) treatments. HPS: High-pressure sodium.

**Figure 2 plants-11-02982-f002:**
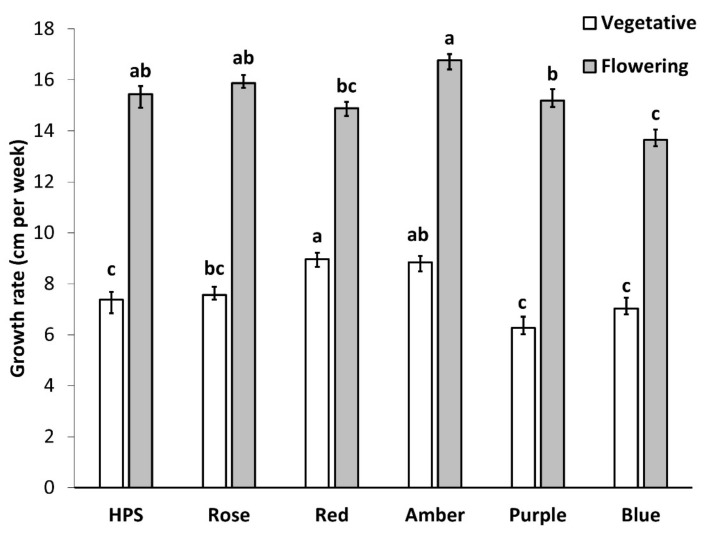
Effect of light spectra on the vertical growth rate of *C. sativa* during the vegetative phase (white) and the first three weeks of the flowering phase (grey) in the ‘Babbas Erkle Cookie’ accession. The results are mean and SE (*n* = 9). Different letters (a, b, c) above the bars represent a significant difference between treatments by Tukey HSD (*p* < 0.05). Treatments are high-pressure sodium (HPS), Blue (430 nm), Red (630 nm), Rose (430 + 630 nm, B:R ratio 1:10), Purple (430 + 630 nm, B:R ratio 2:1), and Amber (595 nm).

**Figure 3 plants-11-02982-f003:**
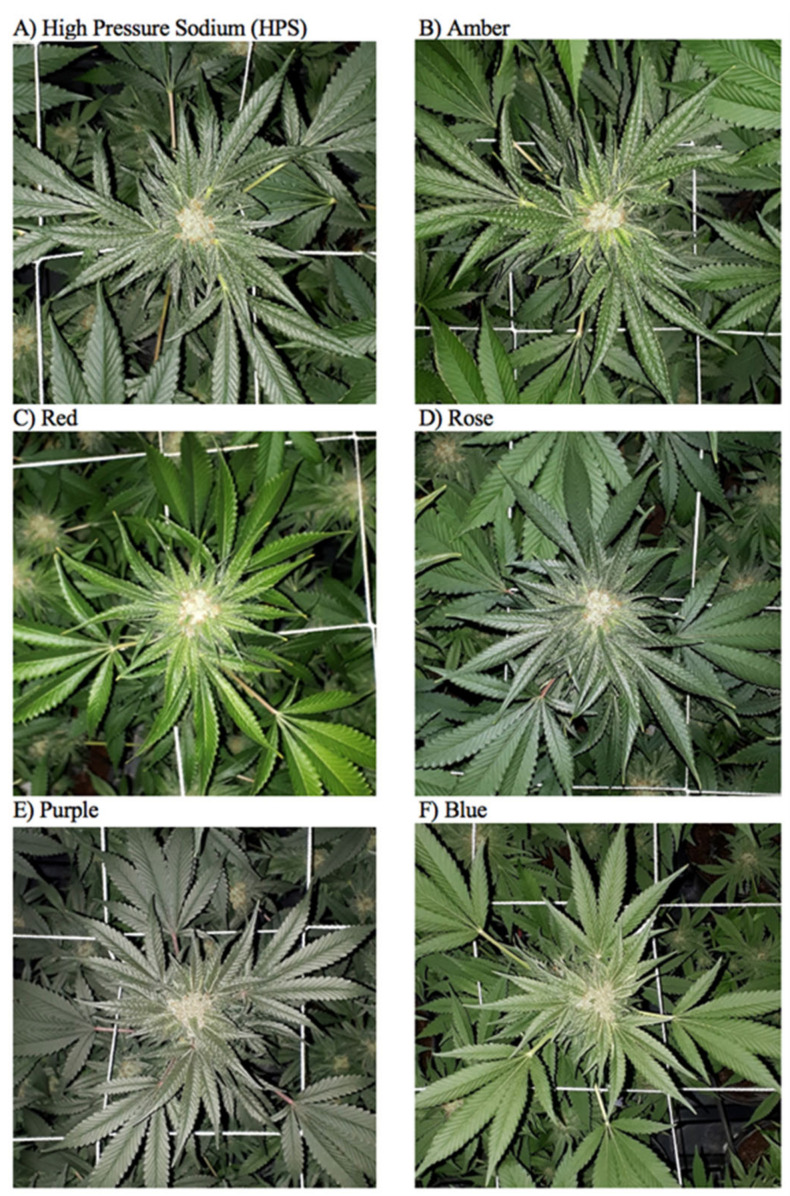
Representative images of leaf and inflorescence coloration under each light treatment one week before harvest. Treatments are (**A**) high-pressure sodium (HPS), (**B**) amber (595 nm), (**C**) red (630 nm), (**D**) rose (430 + 630 nm, B:R ratio 1:10), (**E**) purple (430 + 630 nm, B:R ratio 2:1), and (**F**) blue (430 nm).

**Figure 4 plants-11-02982-f004:**
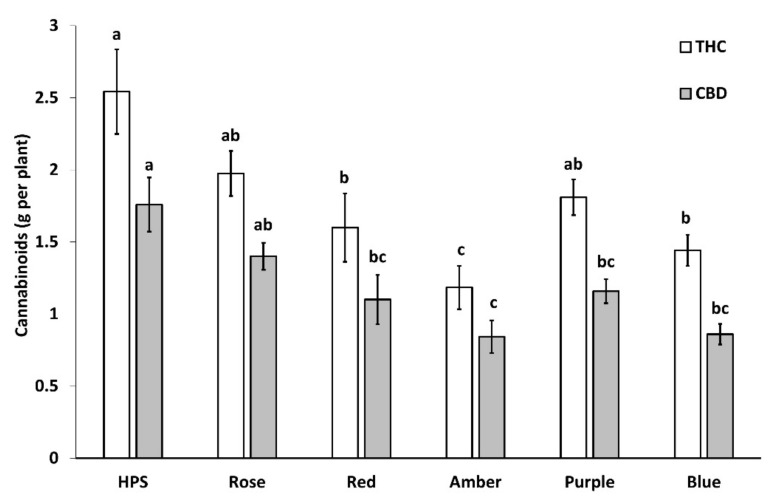
Effect of light treatment on THC and CBD yield per plant. Treatments are high-pressure sodium (HPS), rose (430 + 630 nm, B:R ratio 1:10), red (630 nm), amber (595 nm), purple (430 + 630 nm, B:R ratio 2:1), and blue (430 nm). Values presented in mean and SE (g plant^−1^; *n* = 3). Different letters (a, b, c) above the bars represent a significant difference between treatments by Tukey HSD (*p* < 0.05).

**Figure 5 plants-11-02982-f005:**
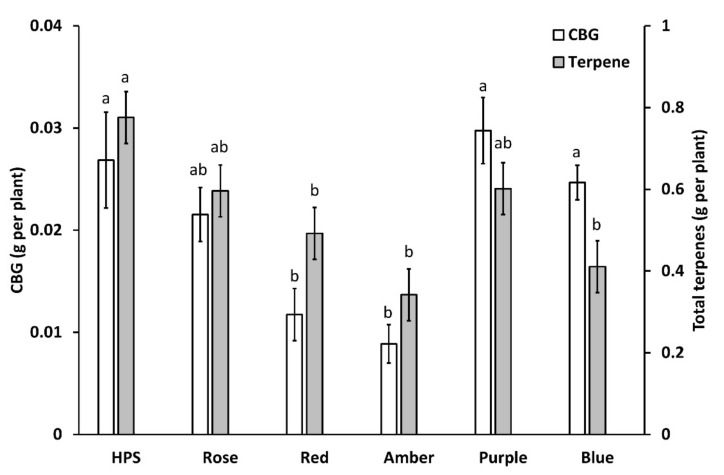
Effect of light treatment on CBG and total terpene yield. Treatments are high-pressure sodium (HPS), rose (430 + 630 nm, B:R ratio 1:10), red (630 nm), amber (595 nm), purple (430 + 630 nm, B:R ratio 2:1), and blue (430 nm). Values presented in mean ± SE (*n* = 3). Different letters (a, b) above the bars represent a significant difference between treatments by Tukey HSD (*p* < 0.05).

**Figure 6 plants-11-02982-f006:**
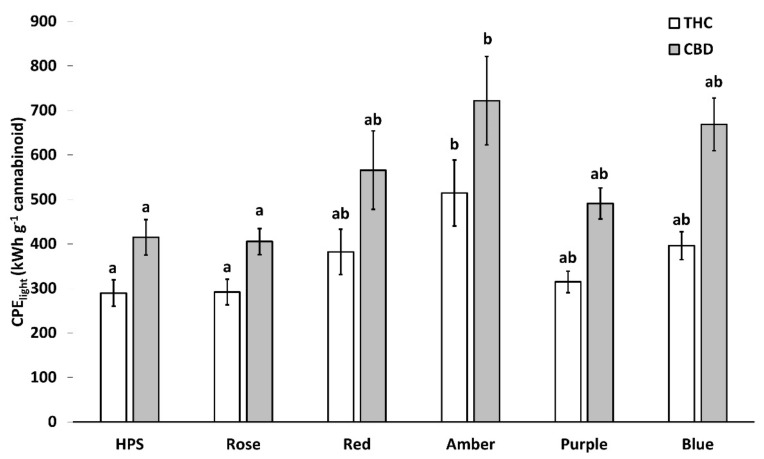
Effect of light treatment on light-associated cannabinoid production efficiency (CPE_light_). Treatments are high-pressure sodium (HPS), blue (430 nm), red (630 nm), rose (430 + 630 nm, B:R ratio 1:10), purple (430 + 630 nm, B:R ratio 2:1), and amber (595 nm). Values presented in mean ± SE (*n* = 3). Different letters (a, b) above the bars represent a significant difference between treatments by Tukey HSD (*p* < 0.05). CPE_ligt_ is given in kWh g^−1^ cannabinoid.

**Table 1 plants-11-02982-t001:** Average final plant height per light treatment per replicate. Values are shown as mean ± SE (*n* = 9). A significant difference was observed between replicates (*p* < 0.0001). Treatments are high-pressure sodium (HPS), Blue (430 nm), Red (630 nm), Rose (430 + 630 nm, B:R ratio 1:10), Purple (430 + 630 nm, B:R ratio 2:1), and Amber (595 nm).

	Average Plant Height at Harvest (cm)	
Light Treatment	Replicate 1	Replicate 2
HPS	78.8 ± 1.6	67.4 ± 2.1
Rose	78.1 ± 1.2	71.2 ± 0.9
Red	76.5 ± 1.2	73.2 ± 1.0
Amber	84.4 ± 1.2	74.8 ± 1.6
Purple	72.0 ± 2.1	69.4 ± 1.2
Blue	68.0 ± 2.1	66.6 ± 1.0

**Table 2 plants-11-02982-t002:** Effect of light treatment on average fresh and dry inflorescence per replicate. Treatments are high-pressure sodium (HPS), Blue (430 nm), Red (630 nm), Rose (430 + 630 nm, ratio B:R 1:10), Purple (430 + 630 nm, B:R ratio 2:1), and Amber (595 nm). The results are mean and SE (*n* = 9).

Light Treatment	Dry Inflorescence (g per Plant)		Fresh Inflorescence (g per Plant)	
	Replicate 1	Replicate 2	Replicate 1	Replicate 2
HPS	28.3 ± 3.5	24.2 ± 7.8	139.3 ± 16.9	127.9 ± 38.7
Rose	27.7 ± 4.1	22.9 ± 3.5	134.8 ± 16.7	123.1 ± 13.6
Red	24.7 ± 3.7	23.0 ± 3.8	126.3 ± 18.1	119.7 ± 17.7
Amber	21.4 ± 3.3	15.3 ± 2.8	112.8 ± 15.9	93.5 ± 20.6
Purple	20.2 ± 4.1	17.3 ± 2.1	101.9 ± 17.4	92.3 ± 9.8
Blue	17.4 ± 3.3	13.5 ± 2.3	82.3 ± 14.9	70.5 ± 9.6

**Table 3 plants-11-02982-t003:** Effect of light treatment on average phytochemical concentration. Treatments are high-pressure sodium (HPS), blue (430 nm), red (630 nm), rose (430 + 630 nm, B:R ratio 1:10), purple (430 + 630 nm, B:R ratio 2:1), and amber (595 nm). Results are shown as mean and SE (*n* = 3). Different letters denote significant differences for total CBG (%) and total terpene (mg g^−1^) (*p* < 0.05).

Light Treatment	Total THC (%)		Total CBD (%)		Total CBG (%)	Total Terpenes (mg g^−1^)
	Replicate 1	Replicate 2	Replicate 1	Replicate 2	Both	Both
HPS	7.7 ± 0.7	9.4 ± 0.8	5.5 ± 0.5	6.3 ± 0.6	0.09 ± 0.03 b	25.8 ± 2.4 ab
Rose	7.0 ± 1.0	9.4 ± 0.2	5.2 ± 0.4	6.4 ± 0.4	0.09 ± 0.03 b	24.5 ± 3.4 ab
Red	5.0 ± 0.5	8.3 ± 1.0	3.6 ± 0.7	5.5 ± 0.9	0.05 ± 0.02 b	20.3 ± 5.3 bc
Amber	5.1 ± 1.8	7.7 ± 0.6	4.0 ± 1.4	5.0 ± 0.5	0.05 ± 0.02 b	18.1 ± 3.2 c
Purple	8.0 ± 0.8	10.0 ± 0.2	5.4 ± 0.5	6.0 ± 0.2	0.15 ± 0.03 a	29.4 ± 2.8 a
Blue	10.0 ± 0.1	10.4 ± 0.3	6.0 ± 0.1	6.1 ± 0.1	0.18 ± 0.03 a	28.5 ± 2.8 a

**Table 4 plants-11-02982-t004:** Concentrations of the six most abundant terpene compounds in *C. sativa* plants cultivated under different light treatments. High-pressure sodium (HPS), Blue (430 nm), Red (630 nm), Rose (430 + 630 nm, B:R ratio 1:10), Purple (430 + 630 nm, B:R ratio 2:1), and Amber (595 nm). Bold numbers are the highest terpene for each light treatment. Each treatment had six biological replicates and values are shown as mean ± SE (*n* = 6). Different letters denote significant differences (*p* < 0.05).

Terpene Compound (mg g^−1^)	Light Treatment					
	HPS	Amber	Red	Rose	Purple	Blue
Myrcene	7.50 ± 0.43 ^ab^	4.80 ± 0.44 ^c^	5.66 ± 0.75 ^bc^	7.34 ± 0.57 ^ab^	8.91 ± 0.53 ^a^	7.89 ± 0.65 ^a^
α-Pinene	5.33 ± 0.45	4.48 ± 0.25	4.73 ± 0.12	5.13 ± 0.31	5.71 ± 0.36	5.73 ± 0.38
Limonene	2.66 ± 0.12 ^ab^	1.85 ± 0.16 ^c^	2.17 ± 0.24 ^bc^	2.59 ± 0.17 ^abc^	3.05 ± 0.11 ^a^	2.65 ± 0.23 ^ab^
β-Pinene	2.47 ± 0.16 ^ab^	1.92 ± 0.10 ^b^	2.03 ± 0.11 ^b^	2.30 ± 0.08 ^ab^	2.70 ± 0.13 ^a^	2.68 ± 0.16 ^a^
Linalool	0.717 ± 0.06 ^ab^	0.428 ± 0.06 ^b^	0.467 ± 0.09 ^ab^	0.59 ± 0.06 ^ab^	0.84 ± 0.04 ^a^	0.84 ± 0.02 ^a^
(E)-β-Ocimene	0.59 ± 0.05 ^abc^	0.347 ± 0.039 ^c^	0.437 ± 0.06 ^bc^	0.625 ± 0.06 ^ab^	0.75 ± 0.06 ^a^	0.631 ± 0.05 ^ab^

**Table 5 plants-11-02982-t005:** Measured spectral and electrical parameters of light fixtures used for each light treatment. The distribution of light intensity is given in a 100 nm wide range. The Blue: red light ratios of 1:10 and 2:1 were supplied by LED.

Light Fraction (nm)	Light Treatments (μmol m^−2^ s^−1^)					
	HPS	Amber (595 nm)	Red (630 nm)	Rose (430 nm + 630 nm; 1:10)	Purple (430 nm + 630 nm; 2:1)	Blue (430 nm)
300–399	0.2	0.5	0.0	4.1	29.4	44.6
400–499	17.8	0.4	0.2	32.1	236.8	352.6
500–599	164.6	258.5	40.9	37.4	15.2	2.6
600–699	137.5	131.9	355.3	323	117.3	0.0
700–799	15.1	6.8	3.1	2.8	1.0	0.0
>800	65.3	0.8	-	-	-	-
sum	398.8	398.9	399.6	399.4	399.7	399.8
Electrical parameters						
Voltage (VAC)	120	120	120	120	120	120
Current (A)	6.15	4.85	4.92	4.88	4.98	4.90
Power (W)	738	582	590.4	585.6	597.6	588

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
