# Peer review of "Light Quality Impacts Vertical Growth Rate, Phytochemical Yield and Cannabinoid Production Efficiency in Cannabis sativa"

_plants, 2022, doi:10.3390/plants11212982_

Round 1

Reviewer 1 Report

The manuscript is within the scope of the journal. Authors provided some discussion of the impact of light source spectra on cannabis growth. This plant is particularly over-studied, but the impact of lighting conditions in past studies has not been well characterized (thus limiting the repeatability or comparability of studies). To prevent the same mistakes, I suggest authors to make some amendments, specifically to the characterization of the independent variable. For example, they should provide absolute spectra (not relative) in Fig. 1. They should also provide the radiometric quantities, such as irradiance falling on the plants. Nominal categories such as “blue light”, “red wavelength,” or light ratios are not precise enough. For further information and limitations of horticultural lighting metrics, authors can refer to:

Pinho, P., Jokinen, K., & Halonen, L. (2012). Horticultural lighting–present and future challenges. Lighting Research & Technology, 44(4), 427-437.

Durmus, D. (2020). Real-time sensing and control of integrative horticultural lighting systems. J, 3(3), 20.

Author Response

The manuscript is within the scope of the journal. Authors provided some discussion of the impact of light source spectra on cannabis growth. This plant is particularly over-studied, but the impact of lighting conditions in past studies has not been well characterized (thus limiting the repeatability or comparability of studies). To prevent the same mistakes, I suggest authors to make some amendments, specifically to the characterization of the independent variable. For example, they should provide absolute spectra (not relative) in Fig. 1. They should also provide the radiometric quantities, such as irradiance falling on the plants. Nominal categories such as “blue light”, “red wavelength,” or light ratios are not precise enough. For further information and limitations of horticultural lighting metrics, authors can refer to:

Pinho, P., Jokinen, K., & Halonen, L. (2012). Horticultural lighting–present and future challenges. Lighting Research & Technology, 44(4), 427-437.

Durmus, D. (2020). Real-time sensing and control of integrative horticultural lighting systems. J, 3(3), 20.

Thank you for your comment. We do want our experiment to be repeatable and comparable with past and future studies. The main goal of this research was to assess whether basic monochromatic and dichromatic LED light fixtures could rival HPS lighting for cannabinoid production. However, after examining the suggested papers by Pinho et al. and Durmus, we think that the degree of detail with respect to lighting efficiency is beyond the scope of our experiment. Furthermore, the lighting fixtures we used are prototypical and their electrical efficiency is not representative of what future LED fixtures efficiency could be. To this end,

we believe we have followed the necessary minimum reporting guidelines for environmental parameters for a greenhouse experiment, as seen here: https://plantmethods.biomedcentral.com/articles/10.1186/s13007-015-0083-5

Furhtermore, this new version of our manuscript now includes more information regarding the nature of the ‘rose’ and ‘purple’ light treatments, including the B:R ratio each time they are mentioned. We also added Table 5, where we were able to put the absolute unit from the light mapping we performed for each 100 nm wavelength range. We also added the values from on-site electrical intensity (amp) needed to power each light fixture.

We want to thank again the reviewer for pointing out the lack of shared information to better compare our results with other studies using LED fixtures.

Reviewer 2 Report

This research aimed to investigate the effects of single or combined LED light spectra on inflorescence yield, and cannabinoids and terpene profiles of Cannabis sativa. The results may provide light receipt to industrial producers of C. sativa. However, the structure of the experiment and the form of the paper are not neat.

1) First of all, please provide reasons/references for choosing the LED used for the treatment in the INTRODUCTION section.

In my opinion, amber (595 nm) light has a wide bandwidth, so it can be interpreted as the effect of Red + Green 1:1 rather than the effect at the monochromatic light of 595 nm.

Also, expressions such as ‘rose’ and ‘purple’ can be confused with monochromatic LEDs, so I think it is better to use the R:B ratio.

2)       And then, please provide the botanical interpretations of these results in the DISCUSSION section, such as plant morphological and phytochemical response to light spectrum.

3)       There are too many materials and methods in the RESULT section. If you wrote according to the journal template, you would rather arrange it in the order of Introduction, M&M, Result, Discussion, and Conclusion.

Methods and results are mixed in one section, so the results are not well read.

4)       In the captions of Tables 1 and 2, ‘Each treatment had 3 biological replicates and values are…’, but why is the statistic a standard error (n = 9)?

If three plants per treatment were sampled and measured, the statistical value should be calculated and expressed as standard deviation (SD) with n = 3.

And in Tables 1 and 2, why was the mean comparison between treatments not performed by post-hoc test?

5)       What is ‘balanced’ B:R ratio? In the text, there are expressions of ‘imbalanced’ or ‘balanced’ B:R ratio. Avoid vague expressions as much as possible, please.

There are a lot of major concerns, so I can’t provide minor suggestions.

In my opinion, this paper should be extensively revised and resubmitted.

Author Response

This research aimed to investigate the effects of single or combined LED light spectra on inflorescence yield, and cannabinoids and terpene profiles of Cannabis sativa. The results may provide light receipt to industrial producers of C. sativa. However, the structure of the experiment and the form of the paper are not neat.

1) First of all, please provide reasons/references for choosing the LED used for the treatment in the INTRODUCTION section.

This is a valid point, thank you. The revised version now includes a section explaining why monochromatic Red and Blue light was added. Additional information and justification for the use of Amber light and its effect on plants was also added. Finally, the reasoning behind the chosen B:R ratio was explained in more detail.

In my opinion, amber (595 nm) light has a wide bandwidth, so it can be interpreted as the effect of Red + Green 1:1 rather than the effect at the monochromatic light of 595 nm.

This is an interesting observation. However, the term amber LED has now been used for more than 23 years. It was originally used to describe a LED having a peak at 595 nm with a 100-150 nm-wide spread (Mukai et al, 1999). A Red+Green light would have to have two-peaks for this nomenclature to be accurate. The authors feel using the suggested term is not needed and would be confusing as the term Amber LED has been used in other articles. Future researchers using the exact same wavelength will be able to more easily compare their work to studies using the same wavelength.

Takashi Mukai et al 1999 Jpn. J. Appl. Phys. 38 3976,

Also, expressions such as ‘rose’ and ‘purple’ can be confused with monochromatic LEDs, so I think it is better to use the R:B ratio.

Thank you for sharing this observation. We have reworked the sections where these wavelengths are mentioned to make sure that monochromatic, dichromatic, narrow- and wide-wavelength descriptors are used when mentioning them. We made sure that all figures’ captions now have the adequate description of what wavelengths are present and at what ratios when using the expression Purple and Rose. We feel that use of simple names for well-known colors are adequate descriptors for understanding which dichromatic LED fixtures were used in each treatment. Furthermore, we believe using a one-word for treatment makes the test easier to read. It also alleviates the figures.

2)       And then, please provide the botanical interpretations of these results in the DISCUSSION section, such as plant morphological and phytochemical response to light spectrum.

A section highlighting the potential implication of specific molecular pathways in dictating cannabis growth has been added.

3)       There are too many materials and methods in the RESULT section. If you wrote according to the journal template, you would rather arrange it in the order of Introduction, M&M, Result, Discussion, and Conclusion.

Methods and results are mixed in one section, so the results are not well read.

Thank you for sharing your concern. The entire section 2 has been reworked to provide only results, and all the M&M info was moved to section 4 where it belongs.  We would like to thank you again for making us rework these sections. In doing so, it was brought to our attention that the reference numbers were misplaced. The entire reference section has been reworked.

4)       In the captions of Tables 1 and 2, ‘Each treatment had 3 biological replicates and values are…’, but why is the statistic a standard error (n = 9)?

Thank you for your input. Indeed, sample size was lacking from the table 1 caption. It had been added.

If three plants per treatment were sampled and measured, the statistical value should be calculated and expressed as standard deviation (SD) with n = 3.

Our study used 9 plants per treatment per replicate for the morphological measurement (height and weight). From this group, a sub-group of three plants per treatment was analyzed for its phytochemical content, which is why n=3 appears only for phytochemical related figures. Tables 1 and 2 have been revised to indicate sample size. We believe keeping SE as the mean uncertainty descriptor for a sample size of 9 is adequate, as other experts in the field, such as the Bernstein group, are using this exact measurement.

And in Tables 1 and 2, why was the mean comparison between treatments not performed by post-hoc test?

A post-hoc test was not used because there was a significant difference in final height between replicate, the assumption of data independency was violated, which renders post-hoc tests not possible.  For this reason, our team believed it was more adequate to use descriptive statistics for traits that had significant differences between replicates as shown in supplementary table 1. This is why bar-graph figures are only used when the analyzed trait has no significant difference between replicates.

5)       What is ‘balanced’ B:R ratio? In the text, there are expressions of ‘imbalanced’ or ‘balanced’ B:R ratio. Avoid vague expressions as much as possible, please.

Thank you for pointing this out. Indeed, vague and subjective descriptions should always be avoided in the context of science. For the sake of making an objective description, we decided to just name the B:R ratio used when we are comparing previous studies at line 335-336. The term ‘’balanced’’ was removed in the abstract section (Line 25).

Round 2

Reviewer 1 Report

I’m glad authors added more information about the independent variables. It is good there at least there are some guidelines for reporting experimental settings in greenhouse studies, but one can argue the guideline itself  is not perfect and needs to be improved. That is how science improves anyway. 

Author Response

Dear Reviewer,

Thank you very much for you comment. We are deeply grateful for your input on characterizing the independent variable more accurately. We hope the additional information will prove useful for the next research to come. 

As for the guidelines in reporting parameters, we do believe these can also be improved. Not only do these guidelines are dated, but the additional information you shared with the suggested references are adequate for energy efficiency study.

We hope we can collaborate in the future to make better guidelines.

Thank you.

Reviewer 2 Report

I think the structure and content of the manuscript have been improved.

As additional minor comments...

- For non-SI units, the reciprocal is not expressed as -1.

As I know, the expression 'per plant' is correct, not 'plant-1'

- Axis titels in figures start with a capital letter.

- In Table 2, 3, first column name is 'Light treatment'.

Please unify the position of 'replicate' in table 3 with table 2.

- Express the unit of CPE light as 'kWh g-1 cannabinoid' in text and figure

- I know that Supplementary Table 2 is just supplementary, but please indicate the meaning of the alphabet and asterisk in the footnote. 

In addition to the comments, please carefully consider and revise the manuscript.

Author Response

Dear reviewer,

Please find the following answers to your comments

- For non-SI units, the reciprocal is not expressed as -1.

As I know, the expression 'per plant' is correct, not 'plant-1'

 This is true, we have modified the use of ‘’plant-1’’ to ‘’per plant’’ in table 2 and all the figures.

- Axis titles in figures start with a capital letter.

 Figure 1 as been corrected based on this observation. Thank you for pointing this out.

- In Table 2, 3, first column name is 'Light treatment'.

This has been modified in table 2 and 3

Please unify the position of 'replicate' in table 3 with table 2.

 This has been addressed in table 3

- Express the unit of CPE light as 'kWh g-1 cannabinoid' in text and figure

Units of CPElight have been modified accordingly

- I know that Supplementary Table 2 is just supplementary, but please indicate the meaning of the alphabet and asterisk in the footnote. 

Thank you for pointing this out. The Supplementary Table 2 legend has been revised. It has come to our attention that the asterisks in the Supplementary Table 2 was to denote an absence of concentration. In order to prevent mixing up the use of the asterisk to denote statistical significance with an absence of data, we have decided to change the ‘’*’’ in supplementary table 2 to ‘’ND’’. In the legend, we have added that ‘’ND’’ stands for ‘’Not detected’’.

In addition to the comments, please carefully consider and revise the manuscript.

Thank you very much for you input, we have reviewed the article in its entirety.